# Gestational Diabetes Mellitus—Recent Literature Review

**DOI:** 10.3390/jcm11195736

**Published:** 2022-09-28

**Authors:** Robert Modzelewski, Magdalena Maria Stefanowicz-Rutkowska, Wojciech Matuszewski, Elżbieta Maria Bandurska-Stankiewicz

**Affiliations:** 1Endocrinology, Diabetology and Internal Medicine Clinic, Department of Internal Medicine, University of Warmia and Mazury in Olsztyn, 10-719 Olsztyn, Poland; 2Department of Endocrinology, Diabetes and Isotope Therapy, Wroclaw Medical University, 50-367 Wroclaw, Poland

**Keywords:** gestational diabetes mellitus, insulin resistance, behavioral treatment

## Abstract

Gestational diabetes mellitus (GDM), which is defined as a state of hyperglycemia that is first recognized during pregnancy, is currently the most common medical complication in pregnancy. GDM affects approximately 15% of pregnancies worldwide, accounting for approximately 18 million births annually. Mothers with GDM are at risk of developing gestational hypertension, pre-eclampsia and termination of pregnancy via Caesarean section. In addition, GDM increases the risk of complications, including cardiovascular disease, obesity and impaired carbohydrate metabolism, leading to the development of type 2 diabetes (T2DM) in both the mother and infant. The increase in the incidence of GDM also leads to a significant economic burden and deserves greater attention and awareness. A deeper understanding of the risk factors and pathogenesis becomes a necessity, with particular emphasis on the influence of SARS-CoV-2 and diagnostics, as well as an effective treatment, which may reduce perinatal and metabolic complications. The primary treatments for GDM are diet and increased exercise. Insulin, glibenclamide and metformin can be used to intensify the treatment. This paper provides an overview of the latest reports on the epidemiology, pathogenesis, diagnosis and treatment of GDM based on the literature.

## 1. Introduction

Gestational diabetes mellitus (GDM) is a state of hyperglycemia (fasting plasma glucose ≥ 5.1 mmol/L, 1 h ≥ 10 mmol/L, 2 h ≥ 8.5 mmol/L during a 75 g oral glucose tolerance test according to IADPSG/WHO criteria) that is first diagnosed during pregnancy [1]. GDM is one of the most common medical complications of pregnancy, and its inadequate treatment can lead to serious adverse health effects for the mother and child [1,2]. According to the latest estimates of the International Diabetes Federation (IDF), GDM affects approximately 14.0% (95% confidence interval: 13.97–14.04%) of pregnancies worldwide, representing approximately 20 million births annually [3]. Mothers with GDM are at risk of developing gestational hypertension, pre-eclampsia and termination of pregnancy via Caesarean section [4]. In addition, GDM increases the risk of complications, including cardiovascular disease, obesity, and impaired carbohydrate metabolism, leading to the development of type 2 diabetes (T2DM) in both mother and infant [5,6,7]. The increase in the incidence of GDM also leads to a significant economic burden and deserves greater attention and awareness [8].

Despite numerous studies, the pathogenesis of GDM remains unclear, and the results obtained so far indicate a complex mechanism of interaction of many genetic, metabolic and environmental factors [9]. The basic methods of treating GDM include an appropriate diet and increased physical activity, and when these are inadequate, pharmacotherapy, usually insulin therapy, is used. In developing countries, such as Brazil, oral hypoglycemic agents are also used, mainly metformin and glibenclamide (glyburide) [10]. The prevention and appropriate treatment of GDM are needed to reduce the morbidity, complications and economic effects of GDM that affect society, households and individuals. Though it is well established that the diagnosis of even mild GDM and treatment with lifestyle recommendations and insulin improves pregnancy outcomes, it is controversial as to which type and regimen of insulin are optimal, and whether oral agents can be used safely and effectively to control glucose levels.

## 2. Aim of the Study

A review of current literature reports on epidemiology, pathogenesis, diagnosis and treatment of GDM.

## 3. Material and Methods

The study presents an analysis of data that are currently available in the literature that concern the epidemiology, pathogenesis, diagnosis and treatment of GDM. The study was based on reviews, original articles and meta-analyses published in English in the last 10 years.

A literature search was conducted from 1 January 2021 to 31 March 2022 using Web of Science, PubMed, EMBASE, Cochrane, Open Grey and Grey Literature Report. MeSH terms, including “gestational diabetes”, “pregnancy induced diabetes”, “hyperglycemia”, “glucose intolerance”, “insulin resistance”, ”prevalence”, “incidence”, “GDM treatment” and “behavioral treatment”, were used alone or in combination.

## 4. Results and Discussion

### 4.1. Epidemiology

The growing problem of overweight and obesity around the world significantly contributes to the steady increase in the incidence of diabetes, including GDM in the population of women of reproductive age [11]. According to the 2019 report by the International Diabetes Federation (IDF), more than approximately 20.4 million women (14.0% of pregnancies) presented with disorders of carbohydrate metabolism, of which approximately 80% was GDM, i.e., about one in six births was affected by gestational diabetes [3]. Table 1 presents the analysis of the geographical distribution of GDM [3,12].

### 4.2. GDM Risk Factors

The incidence of hyperglycemia in pregnancy increases with age. According to Mosses et al., GDM was diagnosed in 6.7% of pregnancies in general, but in 8.5% of women over 30 years of age [13]. Lao et al. showed the highest risk of developing GDM at the ages of 35–39 compared with younger pregnant women (OR 95% CI: 10.85 (7.72–15.25) vs. 2.59 (1.84–3.67)) [14]. These observations were confirmed by IDF data showing the highest percentage of pregnancies with GDM reaching 37% at the ages of 45–49, which was also conditioned by a lower number of pregnancies with an accompanying general higher percentage of diabetes in this population [3]. The delivery of a macrosomic child is another important factor that may increase the risk of both GDM and DM2 by up to 20% [15]. Even after taking into account the age of the woman, pluriparity remains in a linear relationship to the incidence of GDM [16]. GDM in a previous pregnancy increases the risk of reoccurrence by more than six times [17]. In women with a BMI of at least 30 kg/m^2^, the GDM frequency is 12.3%, and in women with first-line relatives that have a history of GDM, it is 11.6%. The combination of these two factors increases the risk of GDM up to 61% of cases [4,18,19]. More than twice the percentage of pregnancies with GDM was observed in women that were previously treated for polycystic ovary syndrome (PCOS) [20]. Recent studies indicated that the prevalence of GDM is related to the season and that GDM prevalence increases during the summer compared with winter [21,22,23]. Moreover, a 50% increase in the incidence of GDM in pregnancies resulting from in vitro fertilization was described [24].

### 4.3. Diagnosing GDM

The decades-long polemic about the diagnosis of GDM has covered two issues: whether to include all pregnant women or only those with risk factors, and whether to use one- or two-stage diagnostic procedures. A GDM diagnosis is only possible if a previous diagnosis of diabetes (i.e., type 1 or type 2 diabetes) had been excluded early in the pregnancy. Screening of only risk groups may result in GDM not being diagnosed in as many as 35–47% of pregnant women, which is certain to affect obstetric results [25]. The results of the Hyperglycemia Adverse Pregnancy Outcome (HAPO) study of 23,316 women gave a clear outcome that elevated glycemia (but below the threshold for overt diabetes mellitus) showed a linear relationship with the occurrence of maternal and neonatal complications expressed as large for gestational age (LGA) endpoints, the frequency of Caesarean sections, neonatal hypoglycemia and the concentration of the umbilical C-peptide [26]. The current criteria for the diagnosis of GDM introduced by The International Association of Diabetes and Pregnancy Study Groups (IADPSG), which were based on the Hyperglycemia and Adverse Pregnancy Outcome (HAPO) results, found a threefold increase in GDM diagnoses, which suggests an earlier underestimation. The HAPO group sought to identify new screening values that would better identify pregnancies at risk for perinatal complications. The HAPO study demonstrated a positive linear relationship between screening glucose values and adverse perinatal outcomes. Moreover, the study authors found that perinatal risks began to increase in women with glucose values that were previously considered “normal” [27,28]. Therefore, nowadays, the basis of GDM diagnostics is the administration of 75 g of glucose between 24 and 28 weeks of pregnancy in all pregnant women without previously diagnosed diabetes. The treatment of even mild forms of glucose intolerance in GDM offers an added benefit, as demonstrated by the Australian Carbohydrate Intolerance Study in Pregnant Women (ACHOIS) and Maternal-Fetal Medicine Units Network (MFMU). It was shown that the frequency of obstetric complications is reduced depending on hyperglycemia and pregnancy weight gain. In the ACHOIS study, the composite endpoint (neonatal death, perinatal injury, hyperbilirubinemia, neonatal hypoglycemia and hyperinsulinemia) was significantly reduced with antihyperglycemic intervention, and there was also a lower weight gain (by 1.7 kg on average) and a lower incidence of LGA. In the MFMU study, no changes were noted in the composite endpoint, but the incidence of LGA and shoulder dystocia decreased significantly [2,29,30]. The results of these studies showed that most scientific societies implement the recommendations of the IADPSG from 2010 and WHO from 2013 into their daily practice. The introduction of the IADPSG criteria for the screening of GDM increased the prevalence by threefold, albeit with no substantial improvements in GDM-related events for women without risk factors except for reduced risks for LGA, neonatal hypoglycemia and preterm birth [31]. This led to further research on a group of patients with GDM. In a large randomized trial (among 23,792 pregnant women), Hillier et al. showed that one-step screening, as compared with two-step screening, doubled the incidence of the diagnosis of GDM, but did not affect the risks of LGA, adverse perinatal outcomes, primary Caesarean section, or gestational hypertension or pre-eclampsia [32]. The GEMS Trial assessed two diagnostic thresholds for GDM—namely, the currently used, higher diagnostic criteria and the IADPSG, lower diagnostic criteria—for their effects on fetal growth, perinatal morbidity, maternal physical and psychological morbidity, and health service utilization. The recently published results of the GEMS Trial showed that lower glycemic criteria (fasting plasma glucose level of at least 92 mg/dL, a 1 h level of at least 180 mg/dL or a 2 h level of at least 153 mg/dL) for the diagnosis of GDM did not result in a lower risk of a large-for-gestational-age infant than the use of higher glycemic criteria (fasting plasma glucose level of at least 99 mg/dL or a 2 h level of at least 162 mg/dL) [33]. This latest study is another important point in the discussion of the best diagnosis method for GDM. Table 2 presents the criteria for the diagnosis of GDM according to different scientific societies.

Many potential markers of GDM occurrence are being described more and more frequently. The greatest hopes are connected with afamine, adiponectin and 1,5-anhydroglucitol [34,35]. Due to the fact that in many countries, prenatal care is provided by gynecologists who can consult other specialists, it seems important to develop predictive models that allow for the identification of women at the highest risk for gestational diabetes in early pregnancy. The Benhalim-2 2020 model, which takes into account interview and biochemical data (propensity score model: history of GDM, FPG, height, triglycerides, age, ethnic origin, first trimester weight, family history of diabetes, HbA1c), showed the highest sensitivity [36].

### 4.4. Pathogenesis of Carbohydrate Metabolism Disorders in Pregnancy

Several factors may be responsible for the occurrence of GDM, the most important of which are insulin resistance and beta cell dysfunction, as well as genetic, environmental and dietary factors.

#### 4.4.1. Insulin Resistance

In the pathogenesis of GDM, as in type 2 diabetes, a key role is played by insulin resistance and decreased insulin secretion relative to the patient’s needs. We observe GDM in both obese and lean women [37]. Insulin resistance induced by pregnancy overlaps with the pre-pregnancy insulin resistance that is already present in obese women, while in lean women, an impaired first phase of insulin secretion is also dominant [38]. Insulin resistance in pregnancy is predisposed by the diabetogenic effect of placental hormones (human placental lactogen (hPL), human placental growth hormone (hPGH), growth hormone (GH), adrenocorticotropic hormone (ACTH), prolactine (PRL), estrogens and gestagens), increased secretion of pro-inflammatory cytokines (tumor necrosis factor alpha (TNF-α), IL-6, resistin and C-reactive protein (CRP)), adiponectin deficiency, hyperleptinemia and central leptin resistance, impaired glucose transport in skeletal muscles, impaired insulin receptor signaling, and decreased expression and abnormal translocation of GLUT-4 to the cell membrane of adipocytes [39,40,41]. An increased secretion of insulin-antagonistic hormones (placental hormones, cortisol) during pregnancy results in an increased insulin resistance, which, at the end of the third trimester, reaches a value similar to full-blown type 2 diabetes [9,42]. Subclinical inflammation in pregnant women as a result of the synthesis of pro-inflammatory cytokines in the placenta and adipose tissue also leads to insulin resistance [43,44]. So far, the effects on the development of insulin resistance due to TNF-α, IL-6 and C-reactive protein have been best studied. Kirwan et al. stated that an increase in insulin resistance, which is characteristic of pregnancy, most strongly correlates with the increase in TNF-α concentration, considering that TNF-α as a marker of insulin resistance during pregnancy [45]. Furthermore, hyperleptinemia in the first weeks of pregnancy is a predictor of the development of gestational diabetes. According to Qui, the determination of the leptin concentration ≥ 31.0 ng/mL in the 13th week of pregnancy causes a 4.7-fold increase in the risk of GDM compared with the risk at the level of leptinemia of ≤14.3 ng/mL. For every 10 ng/mL increase in leptin concentration, the risk of GDM increases by 21% [46]. At the same time, GDM is characterized by elevated concentrations of leptin, which leads to hyperleptinemia [47]. However, pre-pregnancy BMI is a stronger predictor of leptinemia than GDM perse [48]. In women with gestational diabetes, the concentration of adiponectin is lower than in pregnant women without disturbances of carbohydrate metabolism, regardless of their pre-pregnancy BMI [49]. It was shown that a low adiponectin concentration in the first and second trimesters of pregnancy is a predictor of diabetes development in pregnancy [50]. In the Barbour study, a 1.5–2-fold increase in the level of the p85α PI-3-kinase regulatory subunit was found in both the muscle and adipose tissue of obese pregnant and pregnant GDM women compared to obese non-pregnant women. In women with GDM, a 62% increase in the phosphorylation activity of IRS-1 serine residues was found in striated muscle cells compared with the control group of pregnant women without GDM, which points to insulin resistance post-receptor mechanisms [43].

#### 4.4.2. β-Cell Dysfunction

The analysis of insulin secretion disorders in GDM gives inconclusive results. The mechanisms of β-cell hypertrophy and proliferation, resulting in a 300% increase in insulin secretion in the first two trimesters of physiological pregnancy, is insufficient to explain GDM [9,39]. In the pathogenesis of GDM, we also observed the influence of autoimmune and genetic factors, such as the presence of anti-insulin and/or anti-insulin antibodies, which are at risk of developing DM1 and latent autoimmune diabetes in adults (LADA) [51]. In cross-sectional studies, the prevalence of mutations in the gene variants GCK, HNF1A, HNF4A, HNF1B and INS in maturity-onset diabetes of the young (MODY) was 0–5% [52]. Great hopes in the search for the genetic causes of GDM are associated with research on the single nucleotide polymorphism (SNP) related to the cyclin-dependent kinase 5 (CDK5) regulatory subunit associated protein1-like1 gene (CKDAL1). Their presence is associated with an impaired first phase of insulin secretion in DM2 and GDM and leads to a decrease in the mass of beta cells and impairment of their function, leading to GDM [53,54].

#### 4.4.3. Other Factors

A study conducted in Spain showed that carriers of the gene rs7903146 T-allele who followed the Mediterranean diet in early pregnancy had a lower risk of developing GDM [55]. A growing body of research provides evidence of the importance of DNA methylation in the regulation of gene expression associated with metabolic disturbances in pregnant women and in the metabolic programming of the fetus in the setting of GDM-induced hyperglycemia [56,57,58]. In subcutaneous and visceral adipose tissue samples, the insulin receptor mRNA/protein expressions were significantly reduced in women with GDM (*p* < 0.05) [56]. Mothers with GDM displayed a significantly increased global placental DNA methylation (3.22 ± 0.63 vs. 3.00 ± 0.46% (±SD), *p* = 0.013) [57]. Additional light was shed on the pathogenesis of GDM by studies on disorders of the placental proteome, where the placental proteome was altered in pregnant women affected by GDM with large-for-gestational-age (LGA), with at least 37 proteins being differentially expressed to a higher degree (*p* < 0.05) as compared with those with GDM but without LGA [59]. In addition, Khosrowbeygi et al. showed that women with GDM had higher values of TNF-α (225.08 ± 27.35 vs. 115.68 ± 12.64 pg/mL, *p* < 0.001) and lower values of adiponectin (4.50 ± 0.38 vs. 6.37 ± 0.59 µg/mL, *p* = 0.003) and the adiponectin/TNF-α ratio (4.31 ± 0.05 vs. 4.80 ± 0.07, *p* < 0.001) than normal pregnant women. The ratio of adiponectin/TNF-α, which decrease significantly in GDM compared with normal pregnancy, might be an informative biomarker for the assessment of pregnant women at high risk of insulin resistance and dyslipidemia and for the diagnosis and therapeutic monitoring aims regarding GDM [60].

### 4.5. COVID-19 Pandemic and GDM

The second severe acute respiratory distress syndrome (SEA) coronavirus (SARS-CoV-2) causes an acute respiratory disease called coronavirus disease 2019 (COVID-19). There are limited data on the impact of SARS-CoV-2 infection on the onset and course of GDM. A living systematic review and meta-analysis of 435 studies reported the incidence of COVID-19 in pregnant women of approximately 10% (7–14%) [61]. The COVID-19 pandemic has caused organizational difficulties related to the correct diagnosis of GDM. In Anglo-Saxon countries, in order to minimize the risk of infection with SARS-CoV-2, replacement of the three-point OGTT was proposed and the assessment of fasting blood glucose and Hba1c were introduced. Postpartum screening postponement and the use of telemedicine were also offered [62]. However, simplifying the diagnosis of GDM in order to avoid the risk of COVID-19 infection was unfortunately associated with the risk of not diagnosing GDM by as much as 20–30%, which may affect obstetric outcomes [63,64,65]. This was confirmed by another study that showed that in the “COVID era”, diagnostics toward GDM cannot be abandoned and the procedures for its detection cannot be simplified [66]. The COVID-19 pandemic increased the incidence of GDM in 2020 compared with 2019 (13.5% vs. 9%, *p* = 0.01), especially in women in the first trimester of pregnancy. Experiencing lockdown during the first trimester of gestation increased the risk of GDM in these women by a factor of 2.29 (*p* = 0.002) compared with women whose pregnancies occurred before and after lockdown [67]. This is undoubtedly influenced by the sedentary lifestyle of women during the pandemic and reduced physical activity, most often caused by the fear of leaving their homes due to COVID-19 [68]. The “lockdown effect” caused a marked deterioration in glycemic control, an increase in the percentage of HBA1c, and weight/BMI gain in patients with DM2 and GDM [69,70].

### 4.6. Treatment of Gestational Diabetes

Regarding women with GDM, due to the lack of randomized clinical trials, it is extremely difficult to propose an unambiguous and uniform model of management in order to achieve obstetric results similar to the population of healthy women. The treatment of GDM is based on consensus and expert opinion. Analyses of Cochrane Database Reviews showed the lack of unambiguous data on the correlation between the intensity of glycemic control and obstetric outcomes [71]. Based on a meta-analysis from 2014–2019, Mitanchez et al. indicated that the greatest impact on reducing the number of obstetric complications is achieved by combining dietary treatment with exercise [72].

#### 4.6.1. Nutritional Treatment

Nutritional recommendations help women to achieve normoglycemia, optimal weight gain and proper development of the fetus, and the introduction of a pharmacological treatment does not release the mother from the obligation to follow the diet [73]. In GDM, it is necessary to develop an individual nutritional plan based on glycemic self-control, optimal weight gain based on pre-pregnancy BMI, and a calculation of energy requirements and macronutrient proportions, as well as taking into account the mother’s nutritional preferences, together with work, rest and exercise [73]. Chao et al. indicated better results when using individualized recommendations for a specific woman with GDM in contrast to general recommendations [74]. It is recommended to eat three main meals and 2–3 snacks a day, often with a snack around 9:30 pm to protect against nocturnal hypoglycemia and morning ketosis [6]. In a prospective observational study using the 24 h online diet and glycemic tool (“Myfood24 GDM”), better glycemic control was demonstrated with more frequent meals [75]. In women with GDM, carbohydrates are the most important macronutrient, and their high consumption can cause hyperglycemia. However, glucose is the main energy substrate of the placenta and fetus, and thus, is necessary for their proper growth and metabolism [76]. According to the ATA, the content of carbohydrates in the diet should constitute 40–50% of the energy requirement, not less than 180 g/day, and consist mainly of starchy foods with a low glycemic index (GI) [6,73]. The recommended dietary fiber intake is 25–28 g per day, which means a portion of about 600 g of fruit and vegetables per day with a minimum of 300 g of vegetables, whole grain bread, pasta and rice [73,77,78]. Protein should constitute about 30% of the caloric value, that is about 1.3 g/kg of b.w./d, with the minimum recommended daily intake of 71 g of protein [73]. Increased intakes of plant protein, lean meat and fish, and reduced intakes of red and processed meats are beneficial in the treatment of GDM and may improve insulin sensitivity [79,80]. A diet with a high fat content is contraindicated (20–30% of the caloric value is recommended, including < 10% saturated fat), as it leads to placental dysfunction and infant obesity, increased inflammation and oxidative stress, and impaired maternal muscle glucose uptake [80,81,82]. The consumption of saturated fat should be limited in favor of the consumption of the polyunsaturated fatty acids (PUFA) n-3 (linolenic acid) and n-6 (linoleic acid), which are the most important fatty acids for fetal growth and development. A total intake of n-3 in the amount of 2.7 g/day is considered safe during pregnancy [77], while additional fish oil supplementation gives inconclusive results [83]. The recommended weight gain in pregnancy amounts to on average 8–12 kg, depending on the initial body weight (Table 3) [78].

A weight gain of over 18 kg is associated with a twice higher risk of macrosomia [84,85]. Many studies show an increase in the need for vitamins and minerals in pregnancy, mainly folic acid, vitamin D and iron. All pregnant women are recommended to supplement daily with 400 µg of folic acid and 5.0 µg of vitamin D; additionally, depending on the dietary intake, 500–900 mg of calcium and 27–40 mg of iron are recommended [77]. The influence of gut microbiota on the development of GDM is interesting [86]. So far, it was shown that in women in the third trimester of pregnancy, GDM was associated with altered intestinal microflora [87]. However, in the conducted studies on the beneficial effects of probiotics in the prevention or treatment of GDM, the results are still inconclusive [88,89,90,91].

The main quality-oriented recommendations include the need to limit or eliminate processed products with a high content of salt, sugar and fats; avoiding unpasteurized milk, raw meat, alcohol and caffeine; and ensuring proper hydration of at least 2 L of water per day. In addition, the effect of the Dietary Approach to Stop Hypertension (DASH) diet on glycemic control was confirmed, and Sarathi et al. indicated that eating a high-protein diet based on soy products reduces insulin requirements in GDM patients [92,93]. Myoinositol (vitamin B8) supplementation or a diet rich in the MYO-INS isomer may improve glycemic control in GDM [94,95].

#### 4.6.2. Exercise in GDM

In women with GDM, the quantitative and qualitative recommendations for exercise are ambiguous in terms of improving glycemic control [96]. Obstetric indications and contraindications should be followed. If there are no contraindications, the available observational studies indicate the safety of physical activity during pregnancy [97]. Activities that can be safely started and continued are walking, cycling, swimming, selected pilates and low-intensity fitness exercises. It is safe to continue with (but not initiate) the following after consulting with one’s obstetrician: yoga, running, tennis, badminton and strength exercises. Pregnant people should avoid contact sports, horse riding, surfing, skiing and diving. The analysis of Aune et al. showed a reduction in the risk of GDM by 38% (RR 0.62, 95% CI 0.41–0.94) in physically active women [98]. An intervention study in overweight patients by Nasiri-Amiri et al. showed a 24% reduction in the risk of GDM in women exercising no more than three times a week [99]. In women with normal body weight, increased physical activity, according to an analysis by Ming et al., resulted in a lower weight gain in pregnancy without affecting the child’s weight or the frequency of Caesarean sections and a 42% reduction in the risk of GDM (RR 0.58, 95% CI 0.37−0.90, *p* = 0.01) [100]. A meta-analysis by Harrison et al. of eight randomized trials showed a significant reduction in fasting and postprandial glucose levels in women with 20–30 min of activity 3–4 times a week [101].

#### 4.6.3. Pharmacological Treatment

Patients who cannot achieve glycemic targets with a properly balanced diet and elimination of dietary errors should be treated pharmacologically [29]. Most studies indicate insulin therapy as the safest form of treatment, and OAD (orally administrated drugs) treatment should be introduced only in the case of the patient’s lack of consent to insulin therapy or its unavailability [102]. Insulin therapy is carried out in the model of functional intensive insulin therapy (FIIT) with the use of subcutaneous injections. The safety of human insulin use in pregnancy was demonstrated [103]. The safety of the use of aspart and detemir analogs was confirmed in randomized trials [104,105,106] and the safety of lispro and glargine analogs was shown in observational studies [107]; none of the studies showed the passage of insulin analogs across the placenta [108,109]. Currently, metformin and glibenclamide are used as oral medications. Metformin and glibenclamide (glyburide) cross the placenta but are unlikely to be teratogenic [110,111]. The metformin in gestational (MiG) diabetes trial was a landmark study; it was one of the largest randomized controlled trials, in which 751 women with GDM prospectively assessed a composite of neonatal complications as the primary outcome and secondary outcome of neonatal anthropometry at birth. It was concluded that metformin alone, or with supplemental insulin, was not associated with increased perinatal complications. This trial was the basis of many subsequent studies to assess the safety and efficacy of metformin use in GDM [112]. Some studies showed that the use of metformin during pregnancy is associated with higher body weight, more visceral and subcutaneous tissue, and higher blood glucose levels when the offspring is 9 years old [113]. The use of glibenclamide, despite its high effectiveness, may result in a higher percentage of intrauterine deaths and neonatal complications, such as hypoglycemia, macrosomia and FGR (fetal growth restriction) [114]. Although there is an increasing amount of evidence that supports the use of glyburide or metformin for GDM, the American Diabetes Association (ADA) and American College of Obstetricians and Gynecologists (ACOG) still recommend insulin as the primary medical treatment if the glycaemic treatment goals are not achieved with lifestyle intervention due to the lack of evidence regarding the long-term safety of the alternatives [115]. Sodium-glucose cotransporter-2 (SGLT2) inhibitors block the transporter located in the proximal tubule of kidneys that promotes renal tubular reabsorption of glucose, which causes a decrease in blood glucose levels due to an increase in renal glucose excretion. Among women with diabetes, UTI during pregnancy can be associated with pyelonephritis and sepsis and potential long-term effects on the neonate [116]. There were some adverse events noted in animal reproductive studies, including adverse effects on renal development when SGLT2 inhibitors were used in the second and third trimesters, although there are no human data available. The use of SGLT2 inhibitors during pregnancy is not recommended [110]. Recently, some studies reported the use of GLP-1 agents in GDM. GLP-1 agents, including dipeptidyl peptidase-4 (DPP-4) inhibitor and glucagon-like peptide-1 receptor agonist (GLP-1 Ra), enhance insulin secretion in pancreatic b-cell and showed many benefits in treating diabetes mellitus type 2 but are not a common choice for GDM [117,118]. In a systematic review that included 516 patients and investigated the use of GLP-1 agents in GDM (at different time points, including the second trimester of pregnancy and after delivery), Chen et al. showed that the use of GLP-1 agents to normalize blood glucose and can improve insulin resistance, as well as reduce the rate of developing postpartum diabetes compared with a placebo. This systematic review suggested that a dipeptidyl peptidase-4 inhibitor and glucagon-like peptide-1 receptor agonist may be beneficial to GDM patients but need rigorously designed clinical trials to demonstrate this. In particular, whether it can be used during pregnancy to improve pregnancy outcomes or better used to prevent developing diabetes after delivery should be investigated [119]. The data of a randomized controlled trial, namely, The Treatment of Booking Gestational Diabetes Mellitus (TOBOGM), compared pregnancy outcomes among women with booking GDM receiving immediate or deferred treatment can provide new insights into the diagnosis and treatment of GDM [120].

## 5. Conclusions

GDM is one of the most common complications of pregnancy and confers lifelong risks to both women and their children. Observational data demonstrated a linear association between maternal glycemic parameters and risks for adverse pregnancy and offspring outcomes. SARS-CoV-2 infection will undoubtedly affect the risk of GDM. Many doubts regarding the diagnostic criteria and treatment of GDM are still under discussion. Treatment with insulin is effective, but costs and patient experiences limit its use in clinical practice. The use of metformin as a first-line agent for GDM remains controversial due to its transplacental passage and limited long-term follow-up data. Further clinical trials are necessary to use other oral hypoglycemic agents to treat GDM. It is very important for patients with GDM to receive behavioral therapy and to closely cooperate with the doctor. Future work in the field should include studies of both clinical and implementation outcomes, examining strategies to improve the quality of care delivered to women with GDM. The screening and treatment for GDM early in pregnancy are very controversial due to the lack of data from large randomized controlled trials. There is an urgent need for well-designed research that can inform decisions on the best practice regarding gestational diabetes mellitus screening and diagnosis.

## Figures and Tables

**Table 1 jcm-11-05736-t001:** The geographical distribution of GDM [3,12].

Occurrence of Gestational Diabetes Mellitus	
Middle East and North Africa (MENA)	27.6% (26.9–28.4%)
Southeast Asia (SEA) (Brunei, Burma, Cambodia, Timor-Leste, Indonesia, Laos, Malaysia, the Philippines, Singapore, Thailand, Vietnam)	20.8% (20.2–21.4%)
Western Pacific (WP)	14.7% (14.7–14.8%)
Africa (AFR)	14.2% (14.0–14.4%)
South America and Central America (SACA)	10.4% (10.1–10.7%)
Europe (EUR)	7.8% (7.2–8.4%)
North America and the Caribbean (NAC)	7.1% (7.0–7.2%)

**Table 2 jcm-11-05736-t002:** The criteria for the diagnosis of GDM according to different scientific societies.

	Fasting	1 h	2 h	3 h	Number of Values for Diagnosis
Criteria	mg/dL (mmol/L)	mg/dL (mmol/L)	mg/dL (mmol/L)	mg/dL (mmol/L)	
ADA/ACOG ^3^ 2003, 2018	95 (5.3)	180 ^1^ (10.0 ^1^)	155 (8.6)	140 (7.8)	2
ADIPS 2014	92 (5.1)	180 (10.0)	153 (8.5)	- (-)	1
DCCPG 2018 ^4^	95 (5.3)	- (10.6)	- (9.0)	- (-)	1
DIPSI 2014 ^5^	- (-)	- (-)	140 (7.8)	- (-)	1
EASD 1991	110 ^1^/126 (6.1 ^1^/7.0)	- (-)	162 ^1^/180 (9.0 ^1^/10.0)	- (-)	1
FIGO 2015	92 (5.1)	180 (10.0)	153 (8.5)	- (-)	1
WHO 1998	110 ^2^/126 (6.1 ^2^/7.0)	- (-)	120 ^2^/140 (6.7 ^2^/7.8)	- (-)	1
WHO 2013	92 (5.1)	180 ^1^ (10.0 ^1^)	153 (8.5)	- (-)	1
IADPSG/WHO	92 (5.1)	180 ^1^ (10.0 ^1^)	153 (8.5)	- (-)	1
NICE	- (5.6)	- (-)	- (7.8)	- (-)	

Notes: ADA—American Diabetes Association, ACOG—American College of Obstetricians and Gynecologists, DCCPG—Diabetes Canada Clinical Practice Guidelines, DIPSI—Diabetes in Pregnancy Society Group India, EASD—European Association for the Study of Diabetes, FIGO—International Federation of Gynecology and Obstetrics, ADIPS—Australasian Diabetes in Pregnancy Society, WHO—World Health Organization, IADPSG—International Association of the Diabetes and Pregnancy Study Groups, NICE—National Institute for Health and Care Excellence. ^1^ There are no established criteria for the diagnosis of diabetes mellitus in pregnancy based on a 1 h post-load value. ^2^ Refers to the whole blood glucose level. ^3^ Recommends either the IADPSG one-step or two-step approach; initial screening by measuring plasma or serum glucose concentration 1 h after a 50 g oral glucose load (GCT). Those exceeding the cut-off perform either a 100 g OGTT or 75 g OGTT, requiring two or more venous plasma concentrations to be met or exceed the threshold. ^4^ Listed in the preferred approach, the alternate approach is the IADPSG, which uses a non-fasting 75 g OGTT. ^5^ Uses a non-fasting 75 g OGTT.

**Table 3 jcm-11-05736-t003:** Weight gain in relation to baseline body weight (BMI).

BMI	Weight Gain in Pregnancy
<18.5 kg/m^2^	12.5–18 kg
18.5–24.9 kg/m^2^	11.5–16 kg
25.0–29.9 kg/m^2^	7–11.5 kg
≥30 kg/m^2^	5–9 kg

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
