# Peer review of "Gestational Diabetes Mellitus—Recent Literature Review"

_jcm, 2022, doi:10.3390/jcm11195736_

Round 1
Reviewer 1 Report
In my opinion, this work lacks a conclusion that would summarize the opinion of the authors regarding the direction of future work and research.
Author Response
We thank you very much for your useful suggestions. Manuscript was corrected according to your suggestions.
Reviewer 2 Report
This review is very important in the field of diabetes and pregnancy, as there are so many doubts regarding diagnosis and treatment. I think you should rewrite it using more recent diagnostic criteria and better specifying each one. Your tables are not numbered and the second has no title. On the use of metformin, there is a large randomized clinical trial that compared metformin versus placebo (Mig Trial).
Author Response

(The authors gave the same response as above.)

Reviewer 3 Report
Thank you very much for allowing me to review your manuscript. It is well-written and contributes significantly to the field of maternal and newborn health.
1. Introduction:
I would suggest adding a diagnostic value for hyperglycemia, e.g., a blood glucose level of 190 mg/dL or 10.6 mmol/L from an initial glucose challenge test.
I would suggest adding a citation/source after the sentence: "Mothers with GDM are at risk of developing gestational hypertension, pre-eclampsia and termination of pregnancy by caesarean section" [Kondracki AJ, Valente MJ, Ibrahimou B, Bursac Z. Risk of large for gestational age births at early, full and late term in relation to pre-pregnancy body mass index: Mediation by gestational diabetes status. Paediatr Perinat Epidemiol. 2021;00:1–11. https://doi.org/10.1111/ppe.12809]
In the sentence "In some countries, oral hypoglycemic agents are also used, mainly metformin and glibenclamide (glyburide), I would mention specifically which countries metformin and glibenclamide are used.
3. Materials and methods:
Are there data sources (Pubmed, Medline, Scopus)? Is there a flow diagram of information through the process of systematic review? Inclusion/exclusion criteria? What type of review? Narrative? Scoping? Systematic? PRISMA statement? Table of summary of included studies? (Source, study design, country)
4.1 Epidemiology
Suggest writng the full name of the organization, i.e., International Diabetes Federation (IDF).
"According to the 2019 report by IDF, about 15.8%of pregnancies presented with disorders of carbohydrate metabolism, i.e. more than 20.4 million women, of which 83.6% was GDM, i.e. about 1 in 6 births was affected by gestational diabetes". I would suggest clarifying/rephrasing this sentence.
Table 1. Southeast Asia. Specifically, what countries are included in this geographical area?
"Lao et al. showed the highest risk of developing GDM at the ag eof 35-40 compared to younger pregnant women (OR 2.63; 95% CI 2.4-2.89)" Could you please clarify this sentence?
In women with a BMI of 30 kg/m2 and higher, the GDM frequency is 12.3%, and in women of first-line relatives with a history of GDM it is 11.6%. The combination of these two factors increases the risk of GDM up to 61% of cases [16]. I would recommend adding the citation/source [Kondracki AJ, Valente MJ, Ibrahimou B, Bursac Z. Risk of large for gestational age births at early, full and late term in relation to pre-pregnancy body mass index: Mediation by gestational diabetes status. Paediatr Perinat Epidemiol. 2021;00:1–11. https://doi.org/10.1111/ppe.12809]
"More than twice the percentage of pregnancies with GDM was observed in women previously treated for PCOS" I would suggest writing out what PCOS stands for: polycystic ovary syndrome"
"Seasonality of GDM incidence in the summer was observed (but this re-quires further research" Unclear what this sentence means. Could you please rephrase it?
"Depression was associated with a 1.54 x higher risk of GDM (OR: 1.54; 95% CI: 1.09-2.17) [20]. A 10-year follow-up of women in the US showed an increase in the incidence of GDM by 3.6% in groups of women who were overweight, low-income, 45-64-year-old and inactive" Suggest deleting this sentence.
"The current criteria for the diagnosis of GDM introduced by The International Association of Diabetes and Pregnancy Study Groups (IADPSG), based on the HAPO results, found even a 3-fold increase in GDM diagnoses, which suggests an earlier underestimation" Could you please discuss the criteria and why was there an underestimation?
"Due to the fact that in many countries care over a pregnant woman is pro-vided by gynecologists referring to other specialists" I would rephrase "care over a pregnant woman" to "prenatal care". Who may refer to other specialists? By identification of risk groups, do the authors mean "identification of women at highest risk for gestational diabetes?" Could the authors expand more on what is the Benhalim-2 2020 model?
4.4. Pathogenesis of carbohydrate metabolism disorders in pregnancy:
I would suggest providing specific examples of factors and I would include a source/citation.
"It has been shown that low adiponectin concentration in the first trimester of pregnancy is a predictor of diabetes development later in pregnancy" What is meant by "later in pregnancy?" The third trimester?
"The mechanisms of β-cell hypertrophy and proliferation, resulting in even a 300% increase in insulin secretion in the first two trimesters of physiological pregnancy, are insufficient in GDM" Rephrase to "is insufficient to explain GDM".
"In the pathogenesis of GDM, we also observe the influence of autoimmune and genetic factors, such as the presence of anti-insulin and/or anti-insulin antibodies, which are at risk of developing DM1 and Latent Autoimmune Diabetes in Adults (LADA) [49]. Maturity Onset Diabetes of the Young 2, 3 and 4 (MODY 2, 3 and 4) are observed in about 5% of GDM cases" Suggest rephrasing to: "In cross-sectional studies, the prevalence of mutations in gene variants GCK, HNF1A, HNF4A, HNF1B, and INS in maturity-onset diabetes of the young (MODY) was 0-5%.[50]"
"A Spanish study showed that carriers of gene rs7903146 T-allele who followed the Mediterranean diet in early pregnancy had a lower risk of developing GDM [53]." Rephrase to "in a study conducted in Spain...."
"A growing body of research provides evidence of the importance of DNA methylation..." How is methylation important in this case? Does it increase or decrease gene expression?
"Additional light on the pathogenesis of GDM is shed by studies on disorders of the placental proteome - the ratio of adiponectin to TNF-alpha as the basis for the occurrence of metabolic disorders and GDM [57]." Is the ratio of adiponectin to TNF-alpha higher or lower? What should the normal ratio be?
4.5 COVID-19 pandemic and GDM
The incidence of COVID-19 in pregnant women is approx. 10% (7-14%) [58]. Rephrase to: "One study from....reported the incidence ...of approximately ....."
"The COVID-19 pandemic increased the incidence of GDM in 2020 compared to 2019, especially in women in the first trimester of pregnancy [64]. This is undoubtedly influenced by the sedentary lifestyle of women during the pandemic and reduced physical activity, most often caused by the fear of leaving home due to COVID-19 [65]" In 2020, during the COVID-19 pandemic the incidence of GDM increased -How much? Especially in the 1st trimester, compared to 2019?
4.6.1. Nutritional treatment
"Nutritional treatment is to provide the woman with normoglycemia, optimal weight gain and proper development of the fetus, and the introduction of pharmacological treatment does not release the mother from the obligation to follow the diet [70]." Rephrase to: "Nutritional recommendations help women achieve...."
4.6.3. Pharmacological treatment
Did the authors also consider newer antidiabetic drugs such as sodium-glucose cotransporter-2 inhibitors (SGLT2i), dipeptidyl peptidase-4 inhibitors (DPP-IVi), and glucagon-like peptide-1 receptor agonists (GLP-1RA) on the treatment of GDM?
Are metformin and glibenclamide teratogenic? Have they been associated with any birth defects? If so, I would recommend that they be mentioned.
A conclusion statement is missing. I would recommend adding a short conclusion paragraph that summarizes the main findings of the above literature review.
Author Response
We thank you very much for your useful suggestions. Manuscript was corrected according to your suggestions. Please see the attachment.

Reviewer 4 Report
This is a literature review of recently published papers dealing with the epidemiology, pathogenesis, diagnosis and treatment of GDM. Although this is a comprehensive review, it does not have the character of a systematic review. The methods and especially the selection of relevant articles should be described in more detail.
Minor comments:
Introduction: „Due to the specificity of the researched group, research into the pathogenesis and treatment of GDM remain a challenge.“ I don't understand what the authors are trying to say…
4.1. Epidemiology: „According to the 2019 report by IDF, about 15.8% of pregnancies presented with disorders of carbohydrate metabolism, i.e. more than 20.4 million women, of which 83.6% was GDM, i.e. about 1 in 6 births was affected by gesta- tional diabetes [3]“ The reference is the same as in the introduction, but the numbers are different.
Table 1. Provide a citation into the legend of the table. Citation No.10 Gestational diabetes mellitus in Europe: prevalence, current screening practice and barriers to screening. Diabet Med 2012;29:844-854. This article refers to the incidence in Europe, but the table also includes incidences in other continents. Can you clarify this?
4.3. Diagnosing GDM. „In the MFMU study no changes were noted in the composite endpoint, but a reduction in the LGA frequency and a smaller (2.2 kg) increase in the weight gain of the pregnant woman were“ The sentence seems incomplete to me.
4.4.1. Insulin resistence. „Also hiperleptinemia…“Should be hyper…
4.6.3. Pharmacological treatment. Please use FGR (fetal growth restriction) instead of IUGR (intrauterine growth restriction).
Author Response

(The authors gave the same response as above.)

Round 2
Reviewer 3 Report
Thank you very much for revising the manuscript according to my suggestions. This manuscript describes an important topic and will attract a wide readership.
Author Response
Thank you very much for your revision.